# Bridging the gap between Performance and Interpretability: An Explainable Disentangled Multimodal Framework for Cancer Survival Prediction

**Aniek Eijpe**[1] 🆔                                                        A.EIJPE@UU.NL

**Soufyan Lakbir**[1,2,3] 🆔                                              S.LAKBIR@UU.NL

**Melis Erdal Cesur**[4] 🆔                                               M.ERDAL@NKI.NL

**Sara P. Oliveira**[4] 🆔                                              S.OLIVEIRA@NKI.NL

**Angelos Chatzimparmpas** [5]🆔                              A.CHATZIMPARMPAS@UU.NL

**Sanne Abeln**[1] 🆔                                                      S.ABELN@UU.NL

**Wilson Silva**[1] 🆔                                         W.J.DOSSANTOSSILVA@UU.NL

[1] *AI Technology for Life, Department of Information and Computing Sciences, Department of Biology, Utrecht University, Utrecht, The Netherlands*

[2] *Department of Metabolic Diseases, Wilhelmina Children's Hospital, University Medical Center Utrecht, Utrecht, The Netherlands*

[3] *Regenerative Medicine Center Utrecht, Utrecht, The Netherlands*

[4] *Computational Pathology, Department of Pathology, The Netherlands Cancer Institute, Amsterdam, The Netherlands*

[5] *Visualization and Graphics, Department of Information and Computing Sciences, Utrecht University, Utrecht, The Netherlands*

## Abstract

While multimodal survival prediction models are increasingly accurate, their complexity limits interpretability, reducing insight into how different data sources drive predictions. We introduce DIMAFx, an explainable multimodal framework that learns disentangled modality-specific and modality-shared representations from histopathology whole-slide images and transcriptomics data. Across multiple cancer cohorts, DIMAFx[1] achieves state-of-the-art performance while improving disentanglement. Through a case-study in breast cancer, we find that the most predictive features are modality-shared, linking solid tumor morphology with late estrogen response—consistent with known biology—while key modality-specific features capture microenvironmental signals from the WSI. These results show that disentangled multimodal representations provide biologically meaningful insights without sacrificing predictive performance, supporting their use in precision medicine.

**Keywords:** Cancer survival prediction, Disentangled representation learning, Multimodal deep learning, Explainable AI, Histopathology whole-slide images, Transcriptomics data

## 1. Introduction

Multimodal models integrating histopathology whole-slide images (WSIs) and transcriptomics data have demonstrated improved cancer survival prediction over unimodal approaches (Li et al., 2022; Song et al., 2024a). However, their complex fusion strategies

---

1. This short paper is a shortened version of our preprint (Eijpe et al., 2026a).

produce opaque representations, limiting interpretability, clinical trust, and biological insight. Disentangled Representation Learning (DRL) offers a promising solution by separating modality-specific and modality-shared information into structured representations (Bengio et al., 2013; Higgins et al., 2018). Building on this, our prior work DIMAF (Eijpe et al., 2026b) showed that state-of-the-art survival prediction is achievable with disentangled representations, yet provided no insight into the biological signals they encode.

To address this, we propose DIMAFx, an extension of DIMAF that embeds the disentangled representations into a unified explainable framework. Concretely, DIMAFx extends DIMAF with enhanced WSI representations, SHAP- (Lundberg and Lee, 2017) and attention-based explainability of both unimodal and multimodal features, and a learnable, disentangled aggregation layer. DIMAFx enables systematic analysis of modality-specific and -shared features, their biological meaning, and their contribution to survival prediction. Across four TCGA cohorts, DIMAFx achieves state-of-the-art survival prediction and improved disentanglement. Through a breast cancer case study, we show that DIMAFx identifies biologically meaningful patterns consistent with known breast cancer biology.

## 2. Methods

DIMAFx predicts cancer survival by integrating H&E-stained WSIs and bulk RNA-Sequencing (RNA-Seq) data. The source code with detailed instructions and hyperparameter configurations is publicly available at https://github.com/Trustworthy-AI-UU-NKI/DIMAFx.

We first construct interpretable unimodal representations from both modalities. The transcriptomic profiles are mapped to 50 curated pathways (Liberzon et al., 2015), which are processed through separate Self-Normalizing Networks (Klambauer et al., 2017) to produce pathway-level features with well-defined biological meaning. WSI patch features are extracted using the pretrained model UNI (Chen et al., 2024) and summarized per slide using a Gaussian Mixture Model (GMM) (Song et al., 2024b). The concatenated mean and mixture weight of each of the 16 GMM components form the WSI features, capturing the morphological pattern of each prototype (annotated by a pathologist) and its prevalence within the slide, respectively. The unimodal representations are fused using the Disentangled Attention Fusion layer (Eijpe et al., 2026b), separately modeling intra- and inter-modal interactions via self- and cross-attention, producing two modality-specific and two modality-shared representations. An attention-based aggregation layer (Ilse et al., 2018) pools each into a single vector, from which a linear predictor computes the final risk score.

The model is optimized using the Cox partial log-likelihood loss (Wong, 1986), and a loss based on Distance Correlation (DC) (Székely et al., 2007; Liu et al., 2021) that disentangles the modality-specific and -shared representations (Eijpe et al., 2026b). We trained the model to predict disease-specific survival (DSS) using 5-fold cross-validation, separately on four TCGA cohorts: BRCA ($n$=928), BLCA ($n$=423), LUAD ($n$=463), and KIRC ($n$=346). Models were trained for 30 epochs using the AdamW optimizer with a batch size of 64.

## 3. Results and Discussion

DIMAFx achieves state-of-the-art survival prediction across the four TCGA cohorts (avg. C-index: 0.708, C-index IPCW: 0.641), comparable to the best performing baselines DI-

MAF (Eijpe et al., 2026b) (C-index: 0.715, C-index IPCW: 0.635) and MMP (Song et al., 2024a) (C-index: 0.700, C-index IPCW: 0.621), and significantly stratifies high- and low-risk patient groups across all cohorts ($p < 0.05$). Moreover, DIMAFx substantially improves disentanglement over DIMAF by 10.1% in DC and achieves comparable orthogonal scores. All averaged results are provided in Table 1 in the Appendix.

To examine the biological information encoded in the disentangled representations, we conducted an in-depth case study on the breast cancer cohort (BRCA). Using SHAP analysis, we identified which modality-shared and modality-specific features contribute most strongly to survival prediction. The most predictive features were modality-shared, including one capturing solid tumor morphology contextualized by transcriptomic signals. Further analysis revealed that this feature is primarily driven by interactions between solid tumor morphology and the late estrogen response pathway. This interaction is shown in Figure 1, where we can see a clear association between this single interaction and the final predicted risk. Specifically, higher-grade morphology is associated with pathway upregulation and elevated predicted risk, consistent with known breast cancer biology (Luo et al., 2022). Among modality-specific features, interacting adipose and stromal morphologies were most influential, capturing microenvironmental signals from the WSIs that are not encoded in the pathway features.

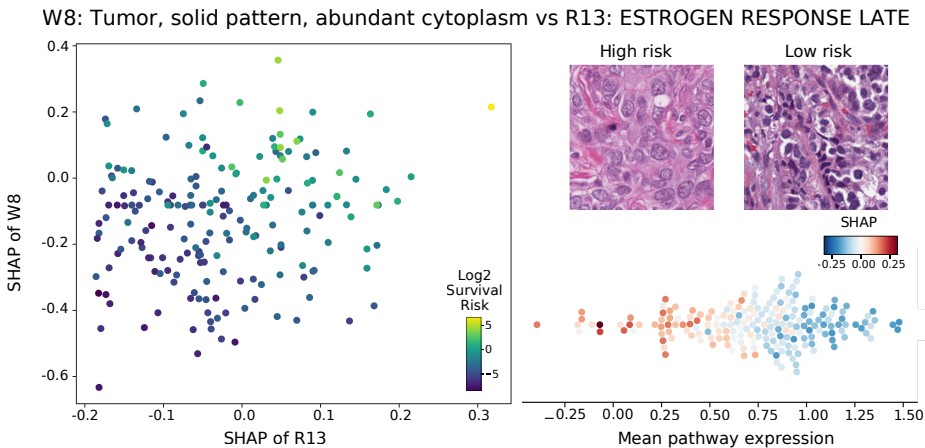

Figure 1: Analysis of a key interaction between the W8 (solid tumor morphological prototype) and R13 (late estrogen response pathway) features identified by DIMAFx.

These results demonstrate that multimodal survival prediction models can achieve strong predictive performance while providing biologically meaningful insights into modality-specific and cross-modal contributions, supporting their application in precision oncology.

## Acknowledgments

This work was supported by the Dutch Research Council (NWO) through the AiNED XS Europa project NGF.1609.241.009.

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

## Appendix A. Full results

Table 1: DSS test results measured by C-index and C-index IPCW averaged across four TCGA cohorts, and disentanglement results measured by Distance Correlation (DC) and orthogonal score (Orth.). The best and second-best performances are denoted by **bold** and underlined, respectively.

| Model | C-index | C-index IPCW | DC | Orth. |
|---|---|---|---|---|
| SurvPath (Jaume et al., 2024) | 0.667 | 0.603 | × | × |
| MMP (Song et al., 2024a) | 0.700 | 0.621 | × | × |
| DIMAFx$_{nodis}$ | 0.700 | 0.626 | 0.766 | **0.058** |
| PIBD (Zhang et al., 2024) | 0.613 | 0.591 | 0.682 | 0.273 |
| DIMAF (Eijpe et al., 2026b) | **0.715** | 0.635 | 0.524 | 0.064 |
| DIMAFx | 0.708 | **0.641** | **0.471** | **0.058** |

