# OpenReview forum: "Bridging the gap between Performance and Interpretability: An Explainable Disentangled Multimodal Framework for Cancer Survival Prediction"
_MIDL.io/2026/Short_Papers — MIDL 2026 - Short Papers Poster_

### Official Review · Reviewer_cnB8 · 2026-05-06
**Review of Bridging the gap between performance and interpretability**

**Rating:** 3
**Confidence:** 4

**Review:**

They find comparable predictive performance to DIMAF and MMP, but better disentanglement of representations. They demonstrate through a case study that the modality-shared information was most predictive and highlight a particular shared feature corresponding to solid tumor morphology and a late estrogen response pathway. This is an interesting concept and an important topic; however, I am not convinced about the novelty of the idea based on the presentation.

**Summary:**

The authors present a framework to learn interpretable “disentangled representations” from multi-modal data where information unique to a modality is separated from information shared among multiple modalities. They extend their prior DIMAF framework for disentanglement with attention-based explainability of unimodal and multimodal features.

They apply their framework to predict cancer survival using RNA-Seq with matched whole-side images from 4 TCGA cohorts. Their approach condenses transcriptomic profiles to 50 curated pathways and WSI features using the UNI model with a GMM to summarize these features per slide.

**Strengths:**

1. Interesting concept to separate modality-shared from modality-specific information. Very helpful to understand multi-modal models
2. Good results in disentanglement and interesting feature case study

**Weaknesses:**

1.	I was not clear about how DIMAF truly differs from DIMAFx. It seems that Shapley values are already integrated into DIMAF. Is the difference just the learnable disentangled aggregation layer? How does this manifest into better explainability?
2.	Like the above comment, were you unable to find W8 x R13 using DIMAF? What is truly novel here?
3.	I wonder if individual case examples of mistaken predictions would be helpful? This might give some insight into whether the explainability can truly help you understand the model’s predictions (most importantly in cases where it is most wrong – high chance of survival but died soon after the WSI or vice versa).
4. Minimal change in c-index for survival prediction with this approach vs. others

**Justification Of Rating:**

Interesting idea, unclear about the novelty

---

### Decision · Program_Chairs · 2026-05-08

Accept (Poster)